# Exploring the strategies and components of interventions to build adolescent awareness about stunting prevention in West Java: A qualitative study

**Nani Nurhaeni**[1][*], **Mega Hasanul Huda**[1], **Siti Chodidjah**[1‡], **Nur Agustini**[1‡], **Fajar Tri Waluyanti**[1‡], **Hartin I. K. Nadi**[1‡], **Ni Ketut Sri Armini**[1‡], **Maya Sari**[1‡], **Debra Jackson**[2‡]

**1** Faculty of Nursing, University of Indonesia, Depok, West Java, Indonesia, **2** London School of Hygiene and Tropical Medicine, London, United Kingdom

☯ These authors contributed equally to this work.
‡ SC, NA, FTW, HIKN, NKSA, MS and DJ also contributed equally to this work.
\* nani-n@ui.ac.id, nanifikui.stuntingproject@gmail.com

**Data Availability Statement:** All relevant data are within the paper.

## Abstract

### Aim

This study aimed to explore the strategies and important components that can be implemented to build adolescent awareness about stunting prevention.

### Methods

This study used descriptive qualitative design. The data were collected through focus group discussions (FGDs) and semi-structured interviews. Purposive sampling method was employed to select the participants. The FGDs involved adolescents (n = 6) and high school counselling teachers (n = 5), while the semi-structured interviews were conducted with experts frequently involved in overcoming stunting problems in Indonesia (n = 7). The interview results were transcribed in verbatim transcription and analysed by using thematic analysis.

### Results

Five themes were identified from the results: 1) Adolescent identity development with three sub-themes: online identity exploration, rebellious stage, and peer influence; 2) Creative and visually appealing website with six sub-themes: interesting appearance, short time span, serial content, story pattern, scenario using adolescent idol's name, and attractive website menu; 3) Nutritional needs for adolescents with three sub-themes: iron and calcium intake, less sugar consumption, and nutritional status; 4) Engaging content for adolescents with seven sub-themes: stunting, reproductive health, anaemia, diet, wellness, early marriage, and physical activity; and 5) Effective communication strategy with two sub-themes: consistency of activities and communicative.

**Funding:** This study was supported by PUTI Grant University of Indonesia NKB- 336/UN2.RST/ HKP.05.00/2023. The funders had no role in study design, data collection and analysis, decision to publish, or preparation of the manuscript.

**Competing interests:** The authors have declared that no competing interests exist.

## Implications

In designing adolescent stunting prevention interventions, multidisciplinary programs utilizing engaging digital health modules and grassroots partnerships should be developed and tested. These programs aim to enhance knowledge retention among youth through appealing online content and interactive community activities. Rigorous evaluation of biopsychosocial approaches can establish integrated best practices across individual, social and policy dimensions for reducing stunting.

## Introduction

Stunting is defined as a child's height-for-age being minus two standard deviations (-2SD) from the reference population median [1]. Global projections in 2019 had anticipated the stunting prevalence among children under five years of age would decline, reaching an estimated 21.3% (144 million). However, the prevalence of stunting rose appreciably in Eastern Africa and Asia by approximately 34.5% and 4.5%, respectively [2]. Prior research has shown malnutrition to be directly or indirectly responsible for 30–50% of mortality in children under five years of age, while stunting alone accounts for around 17% of deaths in this paediatric cohort [3]. According to Indonesia's 2018 Basic Health Survey, the national prevalence of stunting was 29.9% for children under two years and 30.8% for those under five [4]. This prevalence declined to 21.6% by 2022. At the subnational level, the prevalence of stunting in the province of West Java fell from 24.5% to 20.2% over this timeframe [5]. However, the decline has yet to reach Indonesia's national target of reducing stunting below 14% by 2024 [6]. Therefore, efforts to address stunting must continue being optimized, with focused interventions targeting demographic subgroups at highest risk.

Several potential risk factors have been posited as contributing to childhood stunting, including suboptimal maternal nutrition during pregnancy, constrained access to antenatal and paediatric healthcare services, inadequate micronutrient intake, and insufficient health literacy surrounding the preconception, antenatal, and postpartum periods [7–9]. Previous research indicates that mothers experiencing nutritional deficits face an increased risk of fetal disturbances during intrauterine development [10], frequently resulting in infant malnutrition and aberrant postnatal developmental trajectories. Additionally, empirical evidence demonstrates robust associations between parental educational attainment levels and childhood stunting prevalence rates [11], with maternal schooling exerting an even stronger influence.

Stunting prevention may be improved by strengthening adolescents' understanding of preconception nutrition and health, as future parents will be better able to optimize outcomes prior to conception. Therefore, targeting adolescents through early educational interventions aimed at pre-emptively preventing stunting is of strategic importance. However, programs and policies targeting adolescent nutrition are relatively new and coverage remains limited [12]. To meaningfully improve the nutritional status of this vulnerable population, continued evaluation of adolescent-focused programming and strengthened surveillance efforts to capture adolescent nutritional indicators are warranted.

Adolescence (ages 12–18) signifies a developmental transition marked by profound pubertal changes [13]. Recent research finds that 30.17% of youth exhibit insufficient health literacy amid normative transformations [14]. Ongoing cognitive refinement influences optimal information comprehension, while peer and psychosocial factors markedly impact issue

understanding [15]. As adolescents navigate complex health behaviours, tailored approaches are needed to strengthen literacy skills customized to their unique physical, mental, and social trajectories. Health education must also adapt to social and psychological qualities to maximize learning. An evidence-calibrated design for this distinctive stage may optimize literacy and future health via enhanced competence during preparation for adulthood.

School health promotion led by teachers can effectively enhance students' health knowledge and behaviours. Research has shown that educators influence youth development, learning, attitudes, and risks [16]. Teachers are key for health education targeting the development of lifelong healthy practices in students [17]. They also enable open parent-student communication [18]. Active teacher and school involvement thus critically shapes adolescent health understanding. However, lasting behavioural changes require multidimensional prevention coordination across educator and community platforms to optimize consistency in health directives tailored to youth development.

Stunting education for adolescents has been implemented across Indonesia, including programs such as "PENTINGJADI" in West Sumatra [19], educational media in Central Java [20], and a nutrition curriculum in North Sumatra [21]. These aimed to improve youth nutritional knowledge and attitudes to enable early intervention. However, the development of these platforms did not sufficiently consider adolescents' perspectives on preferred learning methods and topics aligned with their developmental stage. Moreover, few initiatives adopted multidisciplinary collaboration despite stunting's multidimensional nature involving nutrition, health, and education. Addressing this requires cross-disciplinary cooperation throughout program design while meaningfully engaging adolescents and educators. Given adolescents' distinctive needs, tailored accommodation is also needed. This study aims to explore effective strategies and important components that can be implemented to build adolescent awareness about stunting prevention.

## Conceptual framework

This conceptual model utilizes Social Learning Theory (SCT) [22] to prevent stunting among adolescents. It focuses on key concepts from SCT, including observational learning [22], self-efficacy [23], behavioural skills [24], outcome expectations, and reinforcement [25] to help youth adopt healthy nutrition behaviours. The model also considers social support [25]. Adolescents will learn positive eating habits by observing nutritionally adept role models in their lives, such as peers without stunting issues [22].

Nutrition education aims to strengthen self-confidence in managing barriers by building skills such as cooking and meal preparation [23]. Having the ability to perform target behaviours increases the likelihood of taking action [23]. Also, teenagers must believe that positive health outcomes, such as growth and development, will result from their actions. Recognition, such as praise and incentives at home and school, further motivates engagement in nutritious diets [25]. Social learning also occurs through guidance and leading by example from caregivers, educators, and community members who create an encouraging environment conducive to optimal eating [24]. This ensures adequate access to diverse, nutrient-dense foods needed for healthy growth and development. Addressing these key concepts from Social Learning Theory empowers adolescents to make informed choices regarding nutrition and prevention of stunting during this important developmental period.

## Methods

This study employed qualitative method to investigate the practices and insights that could be used as the foundation of the research data. In addition, this descriptive qualitative study was

aimed to investigate in-depth information about adolescents' educational needs regarding stunting prevention. We followed the Consolidated Criteria for Reporting Qualitative Research (COREQ) in reporting this study [26]. The interviews represented the part of a larger project to create a website-based educational program for adolescents. The researcher obtained approval from the nursing faculty's ethics committee at the University of Indonesia. Participants filled out the informed consent form as a sign of agreement to participate in the research after receiving an explanation about the study.

## Participants

Eligibility criteria for the current study included experts, adolescents, and counselling teachers who were organized into three participant groups and interviewed using distinct methods. In-depth interviews were conducted with seven key informants comprising two physicians, two paediatric nurses, one psychologist, and two community nutritionists with expertise in fields relevant to stunting prevention. Concurrently, focus group discussions (FGDs) were held involving five counselling teachers with over two years' experience guiding adolescents and six secondary-level students. The sample sizes of teacher and student participants in the focus groups were not expanded beyond five and six individuals respectively. This was because additional data and thematic analysis obtained from FGDs conducted separately with adolescent and teacher groups did not uncover any novel themes relating to perceptions of stunting prevention held by these stakeholders. Thematic saturation was achieved within the current purposively selected sample sizes, suggesting further amplification of participant pools would be unlikely to provide extra conceptual insights. The inclusion of 18 total informants was deemed suitable in accordance with established practices in qualitative methodology. No new information emerged upon completing interviews with the 18th participant. Parents were excluded from involvement in the research due to the ethical consideration that empowering young people to play an active role in shaping their own health and futures necessitates respecting the autonomy and self-determination rights of adolescents [27].

The data was collected under the permission of the Ethical Board of the study site. The experts were chosen using a purposive sampling technique and contacted to ask for their willingness to participate in the interviews. Meanwhile, the counselling teachers and high school students were asked to be involved based on referrals from the key informants using snowball sampling. These participants signed a written consent form, which also included an oral and written explanation of the research objectives. The consent also included permission to record the interviews and FGDs. They were given compensation for participating in this study. No participants refused or dropped out. The entire process took place from August to December 2023. The interviews and FGDs were conducted from 28 August 2023 to 20 December 2023.

## Data collection

Individual interviews and FGDs were conducted to collect the data. Three researchers conducted the individual structured interviews. Meanwhile, one researcher and one observer used an online meeting platform to conduct the FGDs. The data was gathered through online interviews which considered factors like the geographical spread of experts, time efficiency, and cost savings. Prior to the interviews, the researchers looked at challenges with online data gathering such as potential connection issues, ethics guidelines, and ensuring high-quality data [28]. As such, they aimed to help participants by supplying internet allocations, protecting individual privacy, and only using interviewers who were experienced researchers within the field of childhood stunting and qualitative methods. The preparations sought to address the limitations of remote interviewing and yield meaningful findings.

In-depth interviews were conducted individually with each participant using a tailored semi-structured interview guide. The interviews were scheduled asynchronously via separate Zoom rooms to maintain confidentiality and prevent response bias. The two interviewers underwent calibration to ensure consistency in administering questions and facilitating techniques. The interview guide was pretested for construct validity and comprehension, with revisions made as needed. This rigorous yet pragmatic methodology aimed to elicit rich qualitative data in an ethical, standardized manner conducive to credible analysis.

All interviewers had prior expertise in qualitative research. In addition to being lecturers in qualitative research subjects, some interviewers had published qualitative research studies. The interview guide was developed in response to explanatory demands in order to acquire relevant information regarding health education for adolescents, particularly about stunting prevention. The researchers used open-ended questions to capture the essence of experts' and counselling teachers' experiences. Meanwhile, discussions were conducted with the adolescent group, aiming to generate perspective thoughts related to the issue.

## Data analysis

The data collection and analysis were conducted simultaneously and interactively. The earliest part of the analysis began during the interview process, while the data were coded right after the interviews. The interview results were transcribed verbatim, and notes taken during the interviews were used for data analysis. The processes to assess qualitative data included reading the interview results and developing a coding list. The initial coding of the transcripts was conducted independently by the first two authors of the study. When the coding list for each set of participants was created, the next step was to identify similarities and differences in the coding and combine the results from all participants [29]. They then engaged in discussions to determine the final coding framework by combining similar codes and removing duplicate codes. This discussion process yielded a final coding framework that organized the codes into sub-themes and main themes aimed at answering the research questions. Additionally, the other authors discussed the potential themes generated at this stage to ensure they fully captured the data obtained through the research process. Any discrepancies that arose during independent coding were resolved through collaborative discussion between the authors.

In this research, we adopted a thematic analysis approach using both deductive and inductive methods. The deductive approach utilizes an organizing framework comprising predetermined themes to systematically code the data [29–32]. In contrast, the inductive approach involves carefully reading the raw data in detail to derive concepts and themes directly from the content itself without relying on pre-existing constructs. Specifically, this inductive method was implemented by closely examining each line and paragraph of a participant's statement holistically to reveal emerging concepts as the text was read. This bottom-up inductive process differs from the top-down nature of the deductive approach and its reliance on an established theoretical structure to guide analysis.

## Trustworthiness

During the data collection and analysis process, the research team actively participated in iterative peer debriefings to assess team member concerns and preferences. To enhance research validity, member checking was applied where team members actively participated in deliberations pertaining to the findings. A deliberative process was undertaken with the objective of reducing or resolving any disagreements in subsequent instances. Furthermore, the interpretive description was disseminated and discussed among a cohort of five participants in this study. The researchers provided instructions directing participants to engage in introspection

of their personal experiences and to evaluate the extent to which these experiences corresponded with or diverged from the collected findings. Implementing member checking helped align the researchers' conceptual framework with the participants' experiences [33]. Specific quotations and coding were presented to confirm every theme and sub-theme. This study employed methodological triangulation by applying three different data collection approaches: in-depth interviews, observation, and field notes.

## Results

Table 1 summarizes the results of the interviews and FGDs involving 6 adolescents, 5 counselling teachers, and 7 experts. Most of the counseling teachers were female (80%) as well as the high school students, who were also predominantly female (66.7%). Meanwhile, all of the experts involved were female.

From the thematic analysis, this study revealed 5 themes with 24 sub-themes (Table 2). The themes describe the importance of developing education strategies for stunting prevention by considering the unique characteristics of adolescents.

### Adolescent identity development

The participants in the research noted that adolescent identity development needs to be considered in creating educational strategies to avoid stunting in teenagers. Nowadays, young people frequently explore their self-identity online given the simple accessibility of the internet. The participants also stated that adolescents were in their transition period. As Generation Z, teenagers regularly rebel and rely more on their friends. The power and role of their peers are crucial. Therefore, taking advantage of the peer group should be taken into account when designing the intervention.

**Table 1. Demographic characteristic of participants (n = 18).**

| ID | Age | Gender | Job tittle |
|----|-----|--------|------------|
| P1 | | Female | Counselling Teacher |
| P2 | | Female | Counselling Teacher |
| P3 | | Female | Counselling Teacher |
| P4 | | Female | Counselling Teacher |
| P5 | | Male | Counselling Teacher |
| P6 | 16 years | Female | High School Student (10th grade) |
| P7 | 16 years | Female | High School Student 10th grade) |
| P8 | 15 years | Female | High School Student (10th grade) |
| P9 | 15 years | Female | High School Student (10th grade) |
| P10 | 16 years | Male | High School Student (11th grade) |
| P11 | 17 years | Male | High School Student (11th grade) |
| P12 | | Female | Doctor |
| P13 | | Female | Community Nutritionist |
| P14 | | Female | Community Nutritionist |
| P15 | | Female | Psychologist |
| P16 | | Female | Paediatric Nurse |
| P17 | | Female | Paediatric Nurse |
| P18 | | Female | Doctor |

P, Participant.

**Table 2. Identification of themes, sub-themes, the total corresponds to each sub-themes, and the example of each sub-theme.**

| Theme | Sub-theme | Number of quotes (% of theme) | Percentage of total statements | Example (Participant number) |
|---|---|---|---|---|
| 1. Adolescent identity development | | 18 | 9% | |
| | Online identity exploration | 7 (39%) | 4% | P15: ". . .this teenager accesses information that is indeed limited, it is just their same need as their interests and their interests here. . ." |
| | Rebellious stage | 6 (33%) | 3% | P17: ". . .that teenager is a bit difficult to advise, to straighten according to our ideas it is a bit difficult, so sometimes we have to think like teenagers as well. . ." |
| | Peer influence | 5 (28%) | 3% | P7: ". . .. . .In my opinion, health issues that often occur are like smoking, many teenagers nowadays are trying, to smoke/vape, this might occur due to peer influence and lack of family attention. . ." |
| 2. Creative and visually appealing website | | 85 | 44% | |
| | Interesting appearance | 24 (28%) | 13% | P12: ". . ..yes, watching other people's videos is fun, with the movements in the videos and music, you don't have to think right? Unlike leaflets where you have to think and read. . ." |
| | Short time span | 23 (27%) | 12% | P11: ". . ..so nowadays young people are more interested in funny things right, so besides providing information we should also add, intersperse it with some comedy right, it can be through memes or short but dense videos like TikTok for example." |
| | Serial content | 14 (16%) | 7% | P10: "..for example, if there is a lot of information, it can be made into parts one and two. So for each topic, if there are several points, then part one would be one and a half minutes, then part two would be one and a half minutes as well. It can be continued like that. So to put it in terms, it is segmented per section. . ." |
| | Story pattern | 6 (7%) | 3% | P18: Use storytelling patterns for example if using characters so it will depend on the idea later on. |
| | Scenario using adolescent idol's name | 8 (9%) | 4% | P15: ". . .In teenage ages, they are discovering their idols, right. So maybe it's necessary to provide icons—meaning icons taken from current role models or people who are being used as references or role models for teenagers nowadays. First,. . ." |
| | Attractive website menu | 10 (12%) | 5% | P15: ". . .Perhaps there could be an option on the web page for educational games related to the issues or content that we want to deliver." |
| 3. Nutritional needs for adolescents | | 22 | 11% | |
| | Iron and calcium intake | 11 (50%) | 6% | P12: Micronutrient intake still needs attention such as iron, vitamin D and calcium intake; |
| | Less sugar consumption | 5 (23%) | 3% | P18: The health message can start from reducing sugar consumption |
| | Nutritional status (anemia and chronic fatigue syndrom/CFS) | 6 (27%) | 3% | P2: ". . .They were given counseling related to the food composition that must be consumed as the basics in starting their activities in the learning process, so that later they are not lethargic, dizzy, or indeed lack nutritional values in their bodies" |
| 4. Engaging content for adolescent | | 57 | 30% | |
| | Stunting | 14 (25%) | 7% | P18: ". . .In stunting prevention, we must first explain what stunting is, rather than just being short" |
| | Reproductive health | 3 (5%) | 2% | P17: ". . .For teenage friends, it's more about preparing their reproductive health when they get pregnant, then give birth, and then preparations in the early stages because teenagers. . ." |
| | Anemia | 15 (26%) | 8% | P16: "Anemia is, which means an explanation about anemia, what it means, how to prevent it, how to prevent anemia in teenage girls for example is by consuming iron supplements (blood building tablets)." |

*(Continued)*

**Table 2.** (Continued)

| Theme | Sub-theme | Number of quotes (% of theme) | Percentage of total statements | Example (Participant number) |
|---|---|---|---|---|
| | Diet | 11 (19%) | 6% | P12: "...So what is very important is if the diet is wrong, it will also be wrong for what is called growth and development, right?" |
| | Wellness | 4 (7%) | 2% | P12: "One of stunting prevention is to avoid chronic infections, such as tuberculosis. . . . . ." |
| | Early marriage | 6 (11%) | 3% | P17: "...There are many who marry young or take proportions of stunting related to early marriage, what percentage is it? We can explain that, then the impacts." |
| | Physical activity | 4 (7%) | 2% | P15: "..So there are some sports that are currently trending among young people, right? Things like what racing, like running in groups like that, or pound fit, some sports activities, or walking groups.." |
| 5. Effective communication strategy | | 10 | 5% | |
| | Consistency of activities | 2 (20%) | 1% | P12: "Maybe it needs to be continuously emphasized like that sister, and also like this Indonesia is so vast right, so. . ..." |
| | Communicative | 8 (80%) | 4% | P4: "Personally I suggest that the language also needs to be found with a language that is familiar to them, something like the language that young people use nowadays, it shouldn't seem too stiff in language." |

%, percentage.

The form of online identity exploration was supported by the following statements from the participants:

"The generation Z is like that; indeed, they are digital natives, their world is all digital, we see many like that."

(P-15)

"If it is not relatable to their current situation, it will be hard."

(P-12)

". . . I personally suggest to use more familiar language for them, like slangs among young people now, do not make it sound too rigid."

(P-5)

Rebellious stage was supported by these following statements from the participants:

"Middle adolescents are the hardest because of their attitude representing their rebellious stage."

(P-12)

Peer power was supported by these following statements from the participants:

"It is also typical for adolescents to have strong peer relationship, like peer group."

(P-15)

"... and one more from me, actually these kids in their adolescence tend to listen to their friends more..."

(P-2)

"... about the content, maybe like collaborate with content creator, that can relate to their world."

(P-5)

## Creative and visually appealing website

Most of the participants stated that making interesting media should be adjusted to the current situation of adolescents, as they grow along with the rapid development of technology. Interesting appearance was supported by the following statements from the participants:

"... choosing media for these kids to use, like TikTok, oftentimes even my kids here are very familiar with it, like IG."

(P-5)

"... only asking them to read about stunting in a website, it is less interesting, but if they were asked to make content about stunting with movement and writings in the content, it is more interesting for them."

(P-1)

"... maybe like adding some sound effect, like back sound, make it chill, maybe when people are reading, they can also listen to the music, if possible."

(P-11)

"... must be catching, so the website appearance is interesting and simple, but with animation, animation or interesting colours..., colourful, if it is only black and white, reading will be boring, so a little creativity is needed, adding some ornaments to make it interesting."

(P-10)

"... the best way to deliver health education for teenagers, in my opinion, is using video..."

(P-10)

"... using graphics, using images, using animation, using storytelling, it will be more interesting compared to lecturing. Making video using cartoon is also good, just video like infographics, like using PowerPoint, just adding some narration, upload to TikTok, Instagram, and YouTube."

(P-12)

"Mixed method, mixed channel is better than the single one."

(P-14)

"Maybe for early teens, we can make, give more comics, maybe."

(P-15)

"Indeed, audio visual is the best, which means there is short description and pictures that can explain."

(P-17)

"Using gadget, IG live, TikTok."

(P-18)

Teenagers have a fleeting sense of curiosity. They like to switch between different contents to curb feelings of wonder that come and go quickly. In addition, teenagers have a dynamic lifestyle which causes them to engage with social media in between other activities like studying, playing, and interacting. Therefore, creating short-form content is the appropriate choice for developing education for teenagers. Their short attention spans and habits of multitasking between school, entertainment, and socializing online mean that brief videos can effectively raise awareness and convey key messages. Given teenagers' penchant for variety and constant stimulation across various platforms, brief educational clips are well-suited to capturing their interest while imparting important information in a manner respectful of their needs and behaviours. Short time span was supported by the following statements from the participants:

". . . to deliver it, if possible, there are two ways in my opinion, one similar to it, short video like TikTok for example. . ."

(P-11)

"The next is the time span because these kids usually cannot watch those longer than five minutes."

(P-12)

"If using TikTok, it only takes seconds or few minutes finished, but gradually."

(P-13)

"Watching video, four minutes is the longest, four to five minutes."

(P-14)

"So, indeed the audio visual needs to be considered, and also including the duration."

(P-15)

"Maybe right now the phenomenon among the adolescents. . . because of social media, interested in short videos. Video of, if possible, 1 minutes duration, it is long enough, we think of giving education but in their way."

(P-16)

"One minute or one and a half minutes to sixty, right, sixty seconds or one minute to one and a half minutes."

(P-17)

Serial episodes can build a story in a continuous manner which makes teenagers become curious and want to watch the next episode. Series content is also able to make it easier for teenagers to follow the story compared to long episodes. Serial episodes allow the story to be told over multiple shorter instalments, sustaining viewers' interest and desire to find out what

happens next. Stating serial content was supported by the following statements from the participants:

> "Make it in parts, like serial, like Korean drama, it can be long because it was in parts."

> (P-12)

> "So TikTok was made in several chapters for different information."

> (P-13)

> "So, we need to, what is it, make it serial, unless if the one who delivers it is active."

> (P-14)

> "So, the information should indeed be divided into several parts."

> (P-15)

> "Like movies, but in parts"

> (P-16)

The participants explained that education should be developed using story patterns. A clear story pattern has the potential to build an intriguing plot that can capture teenagers' attention. In addition, this pattern will be easier for them to understand and follow. For example, educational development can be carried out by providing illustrations of teenagers who married at a young age. Teenagers' lack of readiness caused them to be unable to prevent stunting in children. By providing realistic examples, teenagers will be able to understand real situations and be able to prevent them. Stating the importance of using story patterns is further supported by evidence showing that framing health messages in an engaging narrative format enhances comprehension and motivation to act compared to a direct didactic approach alone. Stating story patterns was supported by the following statements from the participants:

> "Using story pattern, for example using a character, so it depends on the idea to develop it as serial videos or videos of storytelling."

> (P-12)

> "Video flash future illustrating later when they get married, what will happen, one of which is stunting, and from stunting we can show the suffering."

> (P-16)

> "... the video tells a real story, by giving real example, so we can take it serious... and we can do something to prevent it..."

> (P-10)

The participants felt that using teenage idol names as characters in developing educational content could better capture teenagers' attention. The popularity of idols can facilitate the conveyance of educational messages to teenagers. Scenarios using adolescent idols' names were supported by the following statements:

> "Find current popular names, like Korean names, it is also fine, if those are their idols."

> (P-12)

"Find icons. . . the role model for adolescents, so maybe. . . we need to look for anyone to be invited, like celebgrams or youtubers or else."

(P-15)

"In TikTok, there is one account with short videos. . . even the one who appears in the video is at their age, so they can relate."

(P-16)

The participants suggested developing an effective online health education platform for teenagers that facilitates social interaction. A web-based platform was proposed to enable sharing content on social media and saving links, widening its reach. Gathering feedback from young users through a website comments section was viewed as a way to improve content design and delivery. An attractive website menu was supported by the following statements:

"A place for them to interact like that, to have interaction with those at their age."

(P-15)

"It is like that, so yeah maybe body care, food, or physical activity."

(P-15)

". . . making some content in the form of video game. . . the application, like game, is more suitable. . . have social media, maybe this web-based can be used to save links."

(P-16)

". . . after that maybe we can add games with the theme, so they do not get bored while listening. . ."

(P-6)

"I agree if the web-based becomes the main home, and the contents can be uploaded or published on social media."

(P-17)

". . . the second maybe, ask feedback from the web visitors, maybe we provide a place for them to write their opinion related to the better way to deliver, we collect their feedbacks. . ."

(P-11)

## Nutritional needs for adolescents

The participants explained that it is important to properly address micronutrient intake, such as intake of iron and calcium, to prevent iron deficiency during adolescence. Females who had iron deficiency anaemia as adolescents are more likely to have low birth weight babies and deliver prematurely if they become pregnant. Stating the importance of iron and calcium intake was supported by the following statements from the participants:

"Iron deficiency can be handled by consuming blood supplement tablets, that one of the ways, to obtain higher score and achievement."

(P-12)

"Micronutrient intake should be concerned, like iron, vitamin D, and calcium intake. Iron can prevent iron deficiency anemia, because anemia during adolescence can remain to adulthood. If they get pregnant, they are more likely to have low birth weight and premature baby."

(P-18)

Reducing sugar consumption and limiting carbohydrate-heavy snacks can help address rising diabetes rates. Additionally, reducing intake of sweet beverages like coffee and limiting purchased snacks by instead bringing meals from home can contribute to minimizing sugar and calorie intake. Stating the importance of less sugar consumption was supported by the following statements from the participants:

"Reducing sugar consumption is also possible because most of us can see, maybe twenty or thirty years ago have diabetes."

(P-12)

"They do not want to buy carbo for snacks because we have told them, they bring their own meal from home."

(P-14)

"Reducing coffee and sweet beverages."

(P-18)

The participants also explained that nutritional status is crucial for female adolescents to prepare them to be future mothers. When they become pregnant, adequate iron stores are important. However, many teas that are popular among adolescents can inhibit iron absorption. Additionally, female adolescents often reduce their food portions in pursuit of slimness through dieting, which can be problematic if their diet is improper or taken to an extreme. Clinical issues like anorexia and bulimia may sometimes stem from anxiety over body image wherein girls see themselves as too fat even when underweight. Stating the importance of nutritional status (anaemia and CFS) was supported by the following statements from the participants:

"Nutritional status for female adolescents is aimed to prepare them as future bride and also future mother. . . when they get pregnant, the one in charge is the iron stores."

(P-13)

"The inhibitors of iron absorption are tannin and tea, nowadays many kids like consuming boba tea, Korean tea, and hana tea."

(P-13)

"Female adolescents usually tend to reduce their food portion, for diet, to make them slim."

(P-13)

"So, the very important thing when they do wrong diet."

(P-14)

"It is like anorexia, bulimia, something like that. . . they have anxiety towards their body image, that they are too fat. . . when clinically thin is like that."

(P-15)

## Engaging content for adolescents

The participants stated that the seven contents need to be considered and discussed in designing the intervention for adolescents. The contents are related to one another, including stunting, reproductive health, anaemia, diet, wellness, physical activity, and early marriage. The following statements supported engaging contents for adolescents in preventing stunting:

"In my opinion, it is important, because by knowing stunting, we can put effort to prevent it from happening. . . the same thing for us and the future, if we have children, we know how to manage, to provide good nutrition for the baby. . ."

(P-10)

". . . maybe add the information related to nutritional need to prevent stunting, at least we know the nutrient content of certain food, for example the protein of tempeh, tofu, and egg, maybe like that."

(P-6)

". . . when the students consume the food like that. . . consuming fast food at school. . ."

(P-2)

". . . Using this content, we can give message for them that this is the impact of early marriage, they at this age are very engaged to their physical and social life. . . I think we can use it as the content."

(P-5)

". . . So, the kids can really analyse it for real from the real news, so they know the consequences of promiscuity, the impact for them, for their family, for their future. . ."

(P-1)

". . . for teenagers, it can help them prepare for their reproduction later."

(P-17)

"Many people had early marriage or those took the proportion of stunting because of early marriage."

(P-16)

"And then how they implement parenting, so in terms of quality, this child is better."

(P-17)

"Make sure the food intake is suitable with the recommended dietary allowance (RDA), containing macro and micronutrient."

(P-18)

". . . physical activity."

(P-13)

"do physical activity (aerobic)."

(P-18)

### Effective communication strategy

According to the participants, the way of delivering information to adolescents should use effective communication strategies. The information should be presented gradually and it will be better to use "teenager language" to support the effectiveness of communication and help them understand better. Therefore, the goal can be achieved, especially if it is repeated consistently. Counselling for teenagers was also suggested to support the effectiveness of information transfer. Stating the importance of consistency of activities was supported by the following statements:

"Being conducted repeatedly."

(P-15)

"Conducted together."

(P-15)

Stating communicative was supported by this following statement:

"The way they deliver should be more communicative. . . using adolescent language and using the media that is familiar for teenagers."

(P-17)

Sub-theme 3 stating effective was supported by these following statements:

"The materials focused on 1 issue, for example nutrition, macronutrient, micronutrient, adequate sleep, and physical activity."

(P-18)

". . . involve "friends of their age" while giving the education, we should have a counselling teenager that has been trained to give education for their peers."

(P-18)

## Discussion

### Adolescent identity development

This study is exploratory research that aims to obtain information related to the intervention strategies that can be implemented on adolescents. From the findings, the first theme is adolescent identity development. A previous study explains that understanding the learning need and unique development of adolescent is the key to success in learning process [34]. During adolescence, the physical, emotional, and social transition are developed rapidly. If the

education program is not adjusted to their unique characteristics, it will hinder them from being involved in it [35].

The findings that adolescents spend significant time on social media each day align with past research conducted in Indonesia. Studies by Purboningsih et al. (2022) [36] also found that Indonesian adolescents dedicate more hours to social interaction and browsing online. However, it is important to note that generalizing the exact hours spent online may have limitations depending on regional differences in factors like internet penetration and access to infrastructure within Indonesia. While the influences of digital technologies on enhanced information access and productivity, as highlighted in study of Heidarnia et al. (2016) [37], still hold relevance given the rapid advancement of the digital era, cultural aspects shape adolescents' experiences and behaviours. The study's focus on West Java also means the perspectives may not entirely represent diverse communities across Indonesia's varied geographic, sociocultural, and economic landscapes. Therefore, while digital-based education is appropriate considering participants' screen-based activities, alternative blended approaches could be needed for some rural regions. Additional research exploring potential diversity within the Indonesian adolescent demographic would further strengthen generalization of the findings.

The findings of this study also explain that the unique characteristics of adolescent is rebellious. A Social-Cognitive Theory explained that the authority of a parent to their child is varied in several domains like social, moral, conventional, personal, friendship, and carefulness [38, 39]. Sometimes, most of the children tend to agree that their parents have the authority in several things like moral (no lying and stealing) and things related to health (like smoking, drinking alcohol, and drug abuse). However, for friendship and personal problem, they tend to disobey their parent authority [40]. This finding also supported by a study conducted by Tomé et al. (2012) [41], which explains that parents cannot directly influence the adolescent behaviour, but their friends can significantly do it directly [41]. Social Learning Theory explains that adolescents do not necessarily need to directly watch and adopt a specific behaviour. Instead, perceiving that their peer group approves the behaviour is sufficient for them to consider engaging in comparable behaviours [42]. Therefore, providing health education program involving adolescents and their friends and providing facility for them to adapt are important to be considered by health professionals.

### Creative and visually appealing website

From the analysis, it is found that the form or variation of the website content needs to be developed by health professionals in designing health education for adolescents. This finding is supported by a study indicating that website-based education is effective in reducing smoking behaviour among adolescents in Kermanshah city, Iran [43] and Indonesia [44]. It seems the aesthetic design with interesting colours and clear layout can enhance visual appeal [45]. In addition, responsive design with smooth animation, appealing images, user-friendliness, and easy navigation can also improve positive interaction among users and increase viewer retention [46, 47]. Developing effective website-based health education for adolescents across Indonesia's diverse regions poses challenges due to significant variations in infrastructure access, connectivity, and digital literacy levels between remote, rural, and more populated areas. Thus, a layered approach tailored to regional contexts is needed to address these challenges and maximize the reach and impact of web-based health interventions for adolescents nationwide.

The participants' suggestions regarding the use of engaging short videos and interactive content on websites for health education align with past research on optimal multimedia design preferences for Indonesian adolescents. Study by Alfajri et al. (2014) [48] and Ammerlaan et al. (2015) [49] similarly found that adolescents prefer multimedia elements, games, and

authentic narratives. However, it is important to note that access to technology and levels of digital literacy can vary substantially between Indonesia's diverse regions and demographics based on urban-rural divides. Cultural norms influencing adolescent help-seeking attitudes also differ nationally, so web-based approaches leveraging peer sharing may need localization [50]. Additionally, capturing the viewpoints of a wider range of adolescent subgroups both geographically and demographically throughout Indonesia could provide further contextualization to strengthen the generalization of implications for optimal website design nationally.

In some conditions, however, adolescents encounter challenges in utilizing health information websites when they are excessively text-heavy, visually congested, or possess intricate features that impede accessibility. Involving influencers or public figures famous among teenagers can make the content more relevant and interesting [51]. The use of an idol's name can create identification and affiliation. Teenagers tend to relate more to content involving the figures they adore because it can improve the possibility of sharing the content [52]. It is important to highlight the use of the idols' names to give an extra benefit. Self-identity and affiliation can build a sense of involvement and contribution to create a positive image [53].

## Nutrient needs for adolescents

Calcium and vitamin D also play an important role in bone formation and the healthy growth of the fetus. Adolescents who will become mothers should ensure they have adequate nutrition from food or supplements to prevent the risk of calcium deficiency in babies that might cause stunting [54].

The rate of diabetes as a metabolic disease in adolescents is getting higher. Diabetes also contributes to their health quality as future mothers. Therefore, it is important to address limiting sugar consumption from sources like sweet beverages, fast food, and processed foods as another key topic. Excess sugar intake can lead to issues like obesity and metabolic diseases like diabetes [55]. As a substitute, adolescents should be encouraged to opt for healthy foods with natural sweeteners found in fruits to curb sugar intake. Addressing the health risks of high sugar consumption and promoting low-sugar diet alternatives will help adolescents develop lifestyle habits to support optimal health and nutrition during this critical growth period. Reducing sugar intake is especially important given the rising incidence of diet-related illnesses in young people. Including guidance on limiting added sugars and making healthier sweetener choices provides adolescents with practical knowledge to prevent related conditions that could impact their own health as well as pregnancy outcomes later in life.

From nutritional monitoring, anaemia is commonly caused by iron deficiency. Consuming foods high in iron can help prevent anaemia. Anaemia and chronic fatigue syndrome (CFS) in adolescents are related to imbalanced energy and can be overcome by eating a balanced diet and engaging in adequate physical activity [56]. A balanced diet should include iron-rich foods like meat, fish, poultry, lentils, beans, and leafy greens. Regular physical activity is also important for energy balance and growth during the adolescent period. Together, a nutritious diet and active lifestyle can help adolescents meet their iron needs and prevent fatigue issues associated with anaemia.

## Engaging content for adolescents

There are several key topics that should be addressed in an educational program aimed at preventing stunting among adolescents. The program should provide an explanation of what stunting is, including its definition, causes of stunting such as long-term poor nutrition and frequent infections, and the negative health impacts of stunting [57]. It is also crucial to include material on reproductive health and the increased risk of complications during pregnancy for

stunted girls. The information related to reproductive health services and prenatal care should be understood by female adolescents because these two things will contribute to the fetus's health [58].

Adolescents need to understand the importance of nutrition for growth as well as the risks of anemia, and be taught about elements of a healthy, nutritious diet [59]. Anemia in pregnant women can cause serious health problems. Health education about this issue should contain information related to iron, vitamin, and mineral intake. The right supplements like blood supplement tablets and health monitoring should be prioritized [60].

In addition, the program should promote an overall balanced diet. Adolescents as future mothers should also focus on a balanced diet and healthy lifestyle. The diet should be rich in nutrition, such as fruits, vegetables, and protein, unlike fast food. This lifestyle can improve health and prevent stunting when they become pregnant in the future. Moreover, it is likewise important to discuss the dangers of early marriage and childbearing on female adolescent health and development [61]. In several regions in Indonesia, there are still many early marriages, or marriages before the proper age. Early marriage is one of the contributing factors of stunting. In addition, early marriage also has negative impacts on reproductive health. The impacts of early marriage on mothers' and children's health should be understood by adolescents and the community. Health education through various media should be conducted to give children their right to live healthy and grow optimally [62].

Lastly, the benefits of regular physical activity for bodily development must be covered [63]. Being associated with stunting prevention, fulfilling physical needs, in this case nutritional needs, becomes important. Therefore, addressing all of these topics will best equip adolescents to prevent stunting during this critical period of growth.

## Effective communication strategy

The consistency in providing health education is also identified as a theme in this study. In order to maintain and implement the knowledge they have gained; the activities need to be consistently repeated. These adolescents can become partners in the activities so they can have a sense of belonging. Consistency is a foundation for building understanding and behaviour changes among communities including adolescents. Providing health education that is conducted regularly and consistently is necessary as a stunting prevention effort [64]. This health education can initiate positive habits and support health messages. In communicating directly with teenagers, we need to understand their backgrounds so the communication can be effective. Communicative strategies are also needed to make the information interesting, fit local cultures, and the implementation should use various communication media to make it more efficient.

## Strengths and limitations

This study has many strengths, namely: 1) Data collection was conducted through exploring perceptions from various groups, namely adolescents, counselling teachers, and experts with various experiences in overcoming stunting in Indonesia. The triangulation method was also expected to help the researchers obtain valid information which could be used as materials to design intervention strategies for adolescents; 2) During the FGDs and interviews, an observer was always present to verify the data; 3) The interviews and FGDs were conducted by researchers who were experienced in qualitative studies. In order to improve the quality of this study, verbatim transcriptions were made and discussed among team members.

In addition to its strength, this study also has several limitations that should be taken into account. First, the FGDs and interviews were conducted online. It might cause bias because

the researchers cannot capture the entire response from the participants. Second, during FGDs with the adolescents, it was difficult to have two-way interaction because some of them seemed shy to explain what they wanted to say. Therefore, in the future, offline FGD and interview are more recommended to make the time and place more conducive for the participants and to gain more optimal data. Third, we only included 5 teachers and 6 adolescents in the FGD. Therefore, there is a possibility of bias in the sample selection. Future research should better consider the population size and conduct subject selection by including participants. Fourth, the study has limited external validity due to its narrow sampling from a single Indonesian province, West Java. Further studies investigating possible variations among subgroups within Indonesia's adolescent population could more robustly reinforce the extent to which these initial results may be extended and applied to other similar contexts nationwide.

## Conclusions

Effective adolescent stunting prevention interventions must consider youth characteristics and preferences identified in this formative study. Findings suggest engaging digital modules delivered through accessible websites, alongside attractive, serialized content of judicious length, have potential to improve nutritional knowledge retention. Curricula addressing identified topics like nutrition, diet and physical activity may boost health behaviours. Consistent, communicative online delivery appears key to sustaining comprehension. This preliminary work informs guidelines for developing tested, tailored programs to holistically prevent stunting among adolescents through improved awareness and involvement in nutritious lifestyle promotion.

## Acknowledgments

The authors would like to thank all participants for their contribution to this study.

## Author Contributions

**Conceptualization:** Nani Nurhaeni, Mega Hasanul Huda, Debra Jackson.

**Data curation:** Nur Agustini, Fajar Tri Waluyanti, Ni Ketut Sri Armini, Maya Sari.

**Formal analysis:** Nani Nurhaeni, Siti Chodidjah, Fajar Tri Waluyanti, Hartin I. K. Nadi.

**Investigation:** Ni Ketut Sri Armini.

**Methodology:** Nani Nurhaeni, Mega Hasanul Huda.

**Validation:** Siti Chodidjah, Fajar Tri Waluyanti.

**Writing – original draft:** Nani Nurhaeni, Mega Hasanul Huda, Debra Jackson.

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
