## [Decision Letter · Decision Letter 0]

26 Feb 2024

PONE-D-24-00216Exploring the interventions to build adolescent awareness about stunting prevention: A qualitative studyPLOS ONE

Dear Dr. Nurhaeni,

Thank you for submitting your manuscript to PLOS ONE. After careful consideration, we feel that it has merit but does not fully meet PLOS ONE’s publication criteria as it currently stands. Therefore, we invite you to submit a revised version of the manuscript that addresses the points raised during the review process.

We look forward to receiving your revised manuscript.

Kind regards,

Sana Sadiq Sheikh

Academic Editor

PLOS ONE

“This study was supported by PUTI Grant University of Indonesia NKB- 336/UN2.RST/HKP.05.00/2023.”

“The authors would like to thank all participants for their contribution to this study. This study was supported by PUTI Grant University of Indonesia NKB- 336/UN2.RST/HKP.05.00/2023.”

“This study was supported by PUTI Grant University of Indonesia NKB- 336/UN2.RST/HKP.05.00/2023.”

4. PLOS requires an ORCID iD for the corresponding author in Editorial Manager on papers submitted after December 6th, 2016. Please ensure that you have an ORCID iD and that it is validated in Editorial Manager. To do this, go to ‘Update my Information’ (in the upper left-hand corner of the main menu), and click on the Fetch/Validate link next to the ORCID field. This will take you to the ORCID site and allow you to create a new iD or authenticate a pre-existing iD in Editorial Manager. Please see the following video for instructions on linking an ORCID iD to your Editorial Manager account: https://www.youtube.com/watch?v=_xcclfuvtxQ.

Reviewers' comments:

Reviewer's Responses to Questions

**Comments to the Author**

1. Is the manuscript technically sound, and do the data support the conclusions?

Reviewer #1: Yes

Reviewer #2: Yes

2. Has the statistical analysis been performed appropriately and rigorously? 

Reviewer #1: N/A

Reviewer #2: N/A

3. Have the authors made all data underlying the findings in their manuscript fully available?

Reviewer #1: Yes

Reviewer #2: Yes

4. Is the manuscript presented in an intelligible fashion and written in standard English?

Reviewer #1: Yes

Reviewer #2: Yes

5. Review Comments to the Author

Reviewer #1: The study "Exploring the Interventions to Build Adolescent Awareness about Stunting Prevention" used a descriptive qualitative design to investigate interventions for adolescent stunting prevention. this is an important area to touch upon and may bring policy level implication. The feedback on the methodology is as follows:

Study Participants:

The inclusion of high school counselling teachers aimed to gather insights from professionals directly involved with adolescents. The involvement of experts, including doctors, is essential as they possess specialized knowledge relevant to stunting prevention. While including a community nutritionist could provide valuable insights, the specific expertise of the doctors might have been deemed crucial for the study. Considering parents' perspectives could indeed offer valuable insights into the challenges and opportunities for adolescent stunting prevention, and it would be beneficial to include them in future research.

Online Interviews:

The decision to conduct interviews online might have been influenced by practical considerations such as geographical dispersion of the experts, time efficiency, and cost-effectiveness. However, it's important to acknowledge that online interviews can impact the dynamics of the interaction and the depth of the data collected. Providing a rationale for this choice and addressing any potential implications would strengthen the study.

Findings:

Providing a more detailed explanation of the sub-themes within the identified themes, especially the sub-themes of the first theme, could enhance the understanding of the specific insights gained from the participants.

Whole discussion section is missing, please add that as will further strengthen up your paper.

While the study's methodology has several strengths, such as the use of diverse data collection methods and the involvement of relevant participants, addressing the raised points could further enrich the study's comprehensiveness and rigor.

Reviewer #2: Thank you for the opportunity to review this manuscript explored the interventions that can be implemented to build adolescents’ awareness about stunting prevention. This study used descriptive qualitative design. The data were collected through focus group discussions and semi-structured interviews to adolescents, high school counselling teachers, and experts. Five themes emerged from this study: 1) Adolescent identity development; 2) Creative and visually appealing website; 3) Nutritional needs for adolescents; 4) Engaging content for adolescents; and 5) Effective communication strategy. Overall the authors explore an important research question, and they used appropriate method to answer the research question. I have a few comments that may help authors improve the quality of the manuscript.

1. The authors reported that they used thematic qualitative analysis. Thematic analysis can either be inductive, deductive, or both. In the way it is written it is not clear if they used deductive, or inductive method.

2. Was there any conceptual framework that guided the research study? If so, which framework was used, and how did the authors analyzed data to fit in the domains of the framework? If not, how did the authors develop the themes and sub-themes?

3. It would be great for the authors to describe in the method, what qualitative research guidelines they used ( one example is COREQ).

4. How many people did code the transcripts, and how were coding conflicts resolved?

5. There are a number of spelling mistakes, I would encourage the authors to proof read the manuscript again. For example, the sentences from line 146 – 148 are similar to 152 – 154.

6. In the results, the authors mentioned the themes, sub-themes, and representative quotes. I would recommend the authors use a few sentences to summarize the main message under each theme/ subtheme before stating the quotes. I noted that there are such summaries at the end of each theme, it would be great if these get moved at the beginning, before stating the quotes.

6. PLOS authors have the option to publish the peer review history of their article (what does this mean?). If published, this will include your full peer review and any attached files.

Reviewer #1: **Yes: **Dr. Saleema Gulzar is working as an Associate Professor and Director, Research & Innovation at the Aga Khan University School of Nursing and Midwifery and brings 24 years of experience. In addition, she is serving on an elected board of directors at the eHealth Association of Pakistan. She did her PhD at the University of Sydney, Australia, examining adolescents’ physical activity levels through a mixed-method approach. Her research interest has been towards adolescents’ health through the school health promotion approach, beginning with working as a school health nurse. She is the first in Pakistan to introduce a comprehensive school health promotion program in Pakistan in which her unique contribution was to develop a school health curriculum framework for the schools till the approval of policy for the position of School Nurse at Aga Khan Education Services, Pakistan. She is known for exemplary teaching practice and was awarded for her Research and teaching scholarship. Recently, her teaching has been recognized and selected as member Helie T. Debas teachers’ academy. She is also an active researcher and author of over 70 publications including peer-reviewed papers and book chapters.

Reviewer #2: No

---

## [Author Response · Author response to Decision Letter 0]

17 Apr 2024

Authors reply to the comments from the Associate Editor and Reviewers

Manuscript ID : PONE-D-24-00216

Tittle : Exploring the interventions to build adolescent awareness about 

 stunting prevention: A qualitative study

Chief of Editor comments

1.Please ensure that your manuscript meets PLOS ONE's style requirements, including those for file naming

Response: The authors are most appreciative of the comments from Editor in Chief. We have ensured that our manuscript meets PLOS ONE’s requirements, including those for file naming.

2.Thank you for stating the following financial disclosure:

“This study was supported by PUTI Grant Universitas Indonesia NKB- 336/UN2.RST/HKP.05.00/2023.” 

Response: The authors thank the editor for the comment. We have added the role of the funder in cover letter as follows:

Cover letter: 

3.Thank you for stating the following in the Acknowledgments Section of your manuscript:

“The authors would like to thank all participants for their contribution to this study. This study was supported by PUTI Grant Universitas Indonesia NKB- 336/UN2.RST/HKP.05.00/2023.”

We note that you have provided funding information that is currently declared in your Funding Statement. However, funding information should not appear in the Acknowledgments section or other areas of your manuscript. We will only publish funding information present in the Funding Statement section of the online submission form. Please remove any funding-related text from the manuscript and let us know how you would like to update your Funding Statement. Currently, your Funding Statement reads as follows:

“This study was supported by PUTI Grant Universitas Indonesia NKB- 336/UN2.RST/HKP.05.00/2023.”

Response: Thank you for the comment. We have removed the funding statement from acknowledgement section to submission form. 

4.PLOS requires an ORCID iD for the corresponding author in Editorial Manager on papers submitted after December 6th, 2016. Please ensure that you have an ORCID iD and that it is validated in Editorial Manager. To do this, go to ‘Update my Information’ (in the upper left-hand corner of the main menu), and click on the Fetch/Validate link next to the ORCID field. This will take you to the ORCID site and allow you to create a new iD or authenticate a pre-existing iD in Editorial Manager. Please see the following video for instructions on linking an ORCID iD to your Editorial Manager account: https://www.youtube.com/watch?v=_xcclfuvtxQ

Response: The author thanks for the suggestion from Editor. We have update the information and validate link next to the ORCID field for each number.

5.Please include your full ethics statement in the ‘Methods’ section of your manuscript file. In your statement, please include the full name of the IRB or ethics committee who approved or waived your study, as well as whether or not you obtained informed written or verbal consent. If consent was waived for your study, please include this information in your statement as well.

Response: The authors thank the editor for the suggestion. We have added the information regarding ethics statement including the full name of the IRB and written informed consent in methods section. 

Methods (Page 5, paragraph 1, line 103-106):

The researcher obtained approval from the nursing faculty's ethics committee at the University of Indonesia. Participants filled out the informed consent form as a sign of agreement to participate in the research after receiving an explanation about the study. 

Reviewer 1

The study "Exploring the Interventions to Build Adolescent Awareness about Stunting Prevention" used a descriptive qualitative design to investigate interventions for adolescent stunting prevention. this is an important area to touch upon and may bring policy level implication. The feedback on the methodology is as follows:

1.Study Participants:

The inclusion of high school counselling teachers aimed to gather insights from professionals directly involved with adolescents. The involvement of experts, including doctors, is essential as they possess specialized knowledge relevant to stunting prevention. While including a community nutritionist could provide valuable insights, the specific expertise of the doctors might have been deemed crucial for the study. Considering parents' perspectives could indeed offer valuable insights into the challenges and opportunities for adolescent stunting prevention, and it would be beneficial to include them in future research.

Response: We thank authors for the suggestions. We have added the rationale for not including parents in this research in the methods. 

Methods (Page 5, paragraph 2, line 113-116):

We did not involve parents in the research due to the consideration that if the young generation wants to play an important role in improving their health and become architects of their own future, then the decision-making process must respect the autonomy and choice rights of the young generation (Watson et al., 2023).

2. Online Interviews:

The decision to conduct interviews online might have been influenced by practical considerations such as geographical dispersion of the experts, time efficiency, and cost-effectiveness. However, it's important to acknowledge that online interviews can impact the dynamics of the interaction and the depth of the data collected. Providing a rationale for this choice and addressing any potential implications would strengthen the study.

Response: We thank reviewer for the suggestion. We have provided the rationale for conducting interviews online and addressing any potential implication as well as the strategy for preventing those complications in methods. 

Methods (Page 6, paragraph 2, line 132-139)

Data was gathered through online interviews which considered factors like the geographical spread of experts, time efficiency, and cost savings. Prior to the interviews, researchers looked at challenges with online data gathering such as potential connection issues, ethics guidelines, and ensuring high calibre data (Carter et al., 2021). As such, they aimed to help participants by supplying internet allocations, protecting individual privacy, and only used interviewers who were experienced researchers within the field of childhood stunting and qualitative methods. The preparations sought to address the limitations of remote interviewing and yield meaningful findings.

3. Findings:

Providing a more detailed explanation of the sub-themes within the identified themes, especially the sub-themes of the first theme, could enhance the understanding of the specific insights gained from the participants.

Response: The authors thank the reviewer for the suggestion. We have added the detailed explanation of the sub-themes within the identified themes, particularly for the first theme, by providing the table 2. We also provided the information the total corresponds to each sub-themes, and the example of each sub-theme.

Table 2 (Page 9, line 206-208).

Response: 

4. Whole discussion section is missing, please add that as will further strengthen up your paper.

While the study's methodology has several strengths, such as the use of diverse data collection methods and the involvement of relevant participants, addressing the raised points could further enrich the study's comprehensiveness and rigor.

Response: We have revised the whole discussion in the manuscript. 

Reviewer #2: Thank you for the opportunity to review this manuscript explored the interventions that can be implemented to build adolescents’ awareness about stunting prevention. This study used descriptive qualitative design. The data were collected through focus group discussions and semi-structured interviews to adolescents, high school counselling teachers, and experts. Five themes emerged from this study: 1) Adolescent identity development; 2) Creative and visually appealing website; 3) Nutritional needs for adolescents; 4) Engaging content for adolescents; and 5) Effective communication strategy. Overall the authors explore an important research question, and they used appropriate method to answer the research question. I have a few comments that may help authors improve the quality of the manuscript.

1. The authors reported that they used thematic qualitative analysis. Thematic analysis can either be inductive, deductive, or both. In the way it is written it is not clear if they used deductive, or inductive method.

Response: We thank reviewer for the comment. We used inductive and deductive method of thematic analysis. We have explain that information in methods section.

Methods (Page 7, paragraph 3, line 164-172)

In this research, we adopted thematic analysis by using deductive and inductive approach. The deductive approach utilizes an organizing framework comprising predetermined themes to systematically code the data (Bradley et al., 2007; Braun and Clarke, 2006; Burnard et al., 2008; Miles and Huberman, 1994). In contrast, the inductive approach involves carefully reading the raw data in detail to derive concepts and themes directly from the content itself without relying on pre-existing constructs. Specifically, this inductive method is implemented by the researcher closely examining each line and paragraph of a participant's statement holistically to reveal the emerging concepts as the text is read. This bottom-up inductive process differs from the top-down nature of the deductive approach and its reliance on an established theoretical structure to guide the analysis.

2. Was there any conceptual framework that guided the research study? If so, which framework was used, and how did the authors analysed data to fit in the domains of the framework? If not, how did the authors develop the themes and sub-themes?

Response: We thank reviewer for the comment. We have added the conceptual framework of this research in Conceptual Framework section.

Conceptual Framework (Page 4, Paragraph 2, line 78-94):

This conceptual model utilizes Social Learning Theory (SCT) (Bandura, 1977) to prevent stunting among adolescents. It focuses on key concepts from SCT, including observational learning (Bandura, 1977), self-efficacy (Bandura, 1982), behavioral skills (Baranowski et al., 2003), outcome expectations (Bandura, 1986), reinforcement (Bandura, 1986), and social support (Baranowski et al., 2003) to help youth adopt healthy nutrition behaviors. Adolescents will learn positive eating habits by watching nutritionally adept role models in their lives, such as peers without stunting issues (Bandura, 1977). 

Nutrition education aims to strengthen self-confidence in managing barriers by building skills such as cooking and meal preparation (Bandura, 1982). Having the ability to perform target behaviors increases the likelihood of taking action (Baranowski et al., 2003). Also, teenagers must believe that positive health outcomes, such as development, will result from their actions (Bandura, 1986). Recognition, such as praise and incentives at home and school, further motivates engagement in nutritious diets (Bandura, 1986). Social learning also occurs through guidance and leading by example from caregivers, educators, and community members who create an encouraging environment conducive to optimal eating (Baranowski et al., 2003). This ensures adequate access to diverse, nutrient-dense foods needed for growth. Addressing these key concepts from social learning theory (Bandura, 1977) empowers adolescents to make informed choices regarding nutrition and prevention of stunting during this important developmental period.

3. It would be great for the authors to describe in the method, what qualitative research guidelines they used (one example is COREQ).

Response: We thank reviewer for the comment. We have added the information regarding the guideline in reporting this study on method section.

Methods (Page 5, paragraph 1, line 101-102):

We followed the Consolidated Criteria for Reporting Qualitative Research (COREQ) in reporting this study.

4. How many people did code the transcripts, and how were coding conflicts resolved?

Response: We have added the information regarding the total number of people did code transcript as well as how were coding conflicts resolved in the methods section.

Methods (page 7, paragraph 2, line 154-163):

The initial coding of the transcripts was conducted independently by the first two authors of the study. When the coding list for each set of participants was created, the next step was to identify the similarities and differences in the coding and combined the results from all participants [10]. They then engaged in discussions to determine the final coding framework by combining similar codes and removing duplicate codes. The discussion process yielded a final coding framework that organized the codes into subthemes and main themes aimed at answering the research questions. Additionally, the other authors discussed the potential themes generated at this stage to ensure they fully captured the data obtained through the research process. Any discrepancies that arose during the independent coding were resolved through collaborative discussion between the authors. 

5. There are a number of spelling mistakes, I would encourage the authors to proof read the manuscript again. For example, the sentences from line 146 – 148 are similar to 152 – 154.

Response: We thank the reviewer for the suggestion. We have proof read the manuscript. 

6. In the results, the authors mentioned the themes, sub-themes, and representative quotes. I would recommend the authors use a few sentences to summarize the main message under each theme/ subtheme before stating the quotes. I noted that there are such summaries at the end of each theme, it would be great if these get moved at the beginning, before stating the quotes.

Response: We thank reviewer for the suggestions. We have added a few sentences to summarize the main massage under each theme before stating the quotes for each finding in results section.

Example: Result (Page 11, paragraph 1, line 211-219)

The participants in the research noted that adolescent identity development needs to be considered in creating educational strategies to avoid stunting in teenagers. Nowadays, young people frequently explore their self-identity online given the simple accessibility of the internet. The participants also stated that the adolescents were in their transition period. As a generation Z, teenagers regularly rebel and rely more on their friends. The power and the role of their peers are crucial. Therefore, taking advantage of the peer group should be taken into account in designing the intervention.

---

## [Decision Letter · Decision Letter 1]

27 Aug 2024

PONE-D-24-00216R1Exploring the interventions to build adolescent awareness about stunting prevention: A qualitative studyPLOS ONE

Dear Dr. Nurhaeni,

Thank you for submitting your manuscript to PLOS ONE. After careful consideration, we feel that it has merit but does not fully meet PLOS ONE’s publication criteria as it currently stands. Therefore, we invite you to submit a revised version of the manuscript that addresses the points raised during the review process.

We look forward to receiving your revised manuscript.

Kind regards,

Sana Sadiq Sheikh

Academic Editor

PLOS ONE

Journal Requirements:

Reviewers' comments:

Reviewer's Responses to Questions

**Comments to the Author**

1. If the authors have adequately addressed your comments raised in a previous round of review and you feel that this manuscript is now acceptable for publication, you may indicate that here to bypass the “Comments to the Author” section, enter your conflict of interest statement in the “Confidential to Editor” section, and submit your "Accept" recommendation.

Reviewer #3: (No Response)

Reviewer #4: All comments have been addressed

2. Is the manuscript technically sound, and do the data support the conclusions?

Reviewer #3: Yes

Reviewer #4: Yes

3. Has the statistical analysis been performed appropriately and rigorously? 

Reviewer #3: N/A

Reviewer #4: Yes

4. Have the authors made all data underlying the findings in their manuscript fully available?

Reviewer #3: Yes

Reviewer #4: Yes

5. Is the manuscript presented in an intelligible fashion and written in standard English?

Reviewer #3: Yes

Reviewer #4: Yes

6. Review Comments to the Author

Reviewer #3: Dear Authors,

Your article offers valuable insights into building adolescent awareness about stunting prevention. However, several areas need clarification and enhancement to improve its overall quality.

Title: Since Indonesia is so big and having discrepancy on stunting prevalence among provinces, for accuracy of the title, you can consider to add the place study conducted.

1. Introduction: While the introduction provides a good background on the stunting conceptual framework, it lacks sufficient detail regarding the pivotal factor of teachers, which is also the focus of this research. Consider incorporating primary data, the existing stunting situation in your study place. Providing such information will strengthen the rationale for conducting this research and its contribution to the field of population.

2. Methods: Only 6 adolescent and 5 teacher were interviewed. Can you explain on this. There could have been bias in sampling. We suggest to specify the population size, subject selection methodology, and research process details, including interview conductors and inter-interviewer calibration.

Clarify if interviews were individual and if informants were quarantined. Provide information on the interview guide validation or pilot-testing, and mention any limitations if not validated.

3. Discussion: Compare your findings with existing research in Indonesia and note that the study's focus may limit its generalizability to other regions or countries with different cultural dynamics.

Addressing these points will make your article more comprehensive, transparent, and informative, thereby improving the overall quality of your research.

Reviewer #4: Thanks to excellent improvement, But to improve the quality paper, please fix some items:

1. Abstract: Clarify the specific objective of exploring interventions, summarize the sub-themes mentioned, and strengthen the practical implications with concrete recommendations for future interventions.

2. Introduction: please provide a sharp gap by explicitly stating the specific gap in existing research or interventions your study aims to address.

3. Conclusion: please strengthen for excellence conclusion by refining the language and emphasizing the significance of the findings

7. PLOS authors have the option to publish the peer review history of their article (what does this mean?). If published, this will include your full peer review and any attached files.

Reviewer #3: No

Reviewer #4: **Yes: **Tri Siswati

---

## [Author Response · Author response to Decision Letter 1]

10 Oct 2024

Authors reply to the comments from the Associate Editor and Reviewers

Manuscript ID : PONE-D-24-00216

Tittle : Exploring the strategies and components of interventions to build 

 adolescent awareness about stunting prevention in West Java: A 

 qualitative study

Chief of Editor comments

Response: The author thanks the editor for the suggestions. We have ensured the references throughout the manuscript, both in citations and the reference list.

Reviewer 3:

1. Dear Authors,

Your article offers valuable insights into building adolescent awareness about stunting prevention. However, several areas need clarification and enhancement to improve its overall quality.

Response: The authors truly appreciate the reviewer comments. We will follow the reviewer's suggestions to improve the quality of the manuscript.

2. Tittle: Since Indonesia is so big and having discrepancy on stunting prevalence among provinces, for accuracy of the title, you can consider to add the place study conducted.

Response: We thank the reviewer for their suggestions. We have added the place of the study to the title as follows:

Tittle

Exploring the strategies and components of interventions to build adolescent awareness about stunting prevention in West Java: A qualitative study

3. Introduction: While the introduction provides a good background on the stunting conceptual framework, it lacks sufficient detail regarding the pivotal factor of teachers, which is also the focus of this research. Consider incorporating primary data, the existing stunting situation in your study place. Providing such information will strengthen the rationale for conducting this research and its contribution to the field of population.

Response: The authors thank the reviewer for the comments. We have added the information related to the pivotal factor of teachers in this study and the primary data about prevalence of stunting in the study site in the introduction as follows:

Introduction 

Stunting is defined as a child's height-for-age being minus two standard deviations (-2SD) from the reference population median [1]. Global projections in 2019 had anticipated the stunting prevalence among children under five years of age would decline, reaching an estimated 21.3% (144 million). However, the prevalence of stunting rose appreciably in Eastern Africa and Asia by approximately 34.5% and 4.5%, respectively [2]. Prior research has shown malnutrition to be directly or indirectly responsible for 30-50% of mortality in children under five years of age, while stunting alone accounts for around 17% of deaths in this paediatric cohort [3]. According to Indonesia's 2018 Basic Health Survey, the national prevalence of stunting was 29.9% for children under two years and 30.8% for those under five [4]. This prevalence declined to 21.6% by 2022. At the subnational level, the prevalence of stunting in the province of West Java fell from 24.5% to 20.2% over this timeframe [5]. However, the decline has yet to reach Indonesia's national target of reducing stunting below 14% by 2024 [6]. Therefore, efforts to address stunting must continue being optimized, with focused interventions targeting demographic subgroups at highest risk.

Introduction

School health promotion led by teachers can effectively enhance students' health knowledge and behaviours. Research has shown that educators influence youth development, learning, attitudes, and risks [16]. Teachers are key for health education targeting the development of lifelong healthy practices in students [17]. They also enable open parent-student communication [18]. Active teacher and school involvement thus critically shapes adolescent health understanding. However, lasting behavioural changes require multidimensional prevention coordination across educator and community platforms to optimize consistency in health directives tailored to youth development.

4. Methods: Only 6 adolescent and 5 teacher were interviewed. Can you explain on this. There could have been bias in sampling. We suggest to specify the population size, subject selection methodology, and research process details, including interview conductors and inter-interviewer calibration. Clarify if interviews were individual and if informants were quarantined. Provide information on the interview guide validation or pilot-testing, and mention any limitations if not validated.

Response: We thank the reviewer for their suggestions. In conducting the FGDs, we only included 6 adolescents and 5 teachers without specifying the population size. Due to this reason, we have added this point as one of our limitations. Additionally, we have added information about the rationale for involving 5 teachers and 6 students, the inter-interviewer calibration, as well as the interview guide validation or pilot-testing in the methods section.

Strengths and limitations

In addition to its strength, this study also has several limitations that should be taken into account. First, the FGDs and interviews were conducted online. It might cause bias because the researchers cannot capture the entire response from the participants. Second, during FGDs with the adolescents, it was difficult to have two-way interaction because some of them seemed shy to explain what they wanted to say. Therefore, in the future, offline FGD and interview are more recommended to make the time and place more conducive for the participants and to gain more optimal data. Third, we only included 5 teachers and 6 adolescents in the FGD. Therefore, there is a possibility of bias in the sample selection. Future research should better consider the population size and conduct subject selection by including participants. Fourth, the study has limited external validity due to its narrow sampling from a single Indonesian province, West Java. Further studies investigating possible variations among subgroups within Indonesia's adolescent population could more robustly reinforce the extent to which these initial results may be extended and applied to other similar contexts nationwide.

Methods

Eligibility criteria for the current study included experts, adolescents, and counselling teachers who were organized into three participant groups and interviewed using distinct methods. In-depth interviews were conducted with seven key informants comprising two physicians, two paediatric nurses, one psychologist, and two community nutritionists with expertise in fields relevant to stunting prevention. Concurrently, focus group discussions (FGDs) were held involving five counselling teachers with over two years’ experience guiding adolescents and six secondary-level students. The sample sizes of teacher and student participants in the focus groups were not expanded beyond five and six individuals respectively. This was because additional data and thematic analysis obtained from FGDs conducted separately with adolescent and teacher groups did not uncover any novel themes relating to perceptions of stunting prevention held by these stakeholders. Thematic saturation was achieved within the current purposively selected sample sizes, suggesting further amplification of participant pools would be unlikely to provide extra conceptual insights. The inclusion of 18 total informants was deemed suitable in accordance with established practices in qualitative methodology. No new information emerged upon completing interviews with the 18th participant. Parents were excluded from involvement in the research due to the ethical consideration that empowering young people to play an active role in shaping their own health and futures necessitates respecting the autonomy and self-determination rights of adolescents [27].

Methods

In-depth interviews were conducted individually with each participant using a tailored semi-structured interview guide. The interviews were scheduled asynchronously via separate Zoom rooms to maintain confidentiality and prevent response bias. The two interviewers underwent calibration to ensure consistency in administering questions and facilitating techniques. The interview guide was pretested for construct validity and comprehension, with revisions made as needed. This rigorous yet pragmatic methodology aimed to elicit rich qualitative data in an ethical, standardized manner conducive to credible analysis.

5. Discussion: Compare your findings with existing research in Indonesia and note that the study's focus may limit its generalizability to other regions or countries with different cultural dynamics. Addressing these points will make your article more comprehensive, transparent, and informative, thereby improving the overall quality of your research.

Response: The authors thank the reviewer for the comment. We have strengthened our manuscript by situating our findings within existing Indonesian research and acknowledging limitations in generalizing due to our regional scope in discussion as follows

Discussion

The findings that adolescents spend significant time on social media each day align with past research conducted in Indonesia. Studies by Purboningsih et al. (2022) [36] also found that Indonesian adolescents dedicate more hours to social interaction and browsing online. However, it is important to note that generalizing the exact hours spent online may have limitations depending on regional differences in factors like internet penetration and access to infrastructure within Indonesia. While the influences of digital technologies on enhanced information access and productivity, as highlighted in study of Heidarnia et al. (2016) [37], still hold relevance given the rapid advancement of the digital era, cultural aspects shape adolescents' experiences and behaviours. The study's focus on West Java also means the perspectives may not entirely represent diverse communities across Indonesia's varied geographic, sociocultural, and economic landscapes. Therefore, while digital-based education is appropriate considering participants' screen-based activities, alternative blended approaches could be needed for some rural regions. Additional research exploring potential diversity within the Indonesian adolescent demographic would further strengthen generalization of the findings.

Discussion

From the analysis, it is found that the form or variation of the website content needs to be developed by health professionals in designing health education for adolescents. This finding is supported by a study indicating that website-based education is effective in reducing smoking behaviour among adolescents in Kermanshah city, Iran [43] and Indonesia [44]. It seems the aesthetic design with interesting colours and clear layout can enhance visual appeal [45]. In addition, responsive design with smooth animation, appealing images, user-friendliness, and easy navigation can also improve positive interaction among users and increase viewer retention [46, 47]. Developing effective website-based health education for adolescents across Indonesia's diverse regions poses challenges due to significant variations in infrastructure access, connectivity, and digital literacy levels between remote, rural, and more populated areas. Thus, a layered approach tailored to regional contexts is needed to address these challenges and maximize the reach and impact of web-based health interventions for adolescents nationwide.

Discussion

The participants' suggestions regarding the use of engaging short videos and interactive content on websites for health education align with past research on optimal multimedia design preferences for Indonesian adolescents. Study by Alfajri et al. (2014) [48] and Ammerlaan et al. (2015) [49] similarly found that adolescents prefer multimedia elements, games, and authentic narratives. However, it is important to note that access to technology and levels of digital literacy can vary substantially between Indonesia's diverse regions and demographics based on urban-rural divides. Cultural norms influencing adolescent help-seeking attitudes also differ nationally, so web-based approaches leveraging peer sharing may need localization [50]. Additionally, capturing the viewpoints of a wider range of adolescent subgroups both geographically and demographically throughout Indonesia could provide further contextualization to strengthen the generalization of implications for optimal website design nationally.

Reviewer 4

1. Thanks to excellent improvement, but to improve the quality paper, please fix some items.

Response: Thank you for the insightful feedback on how to strengthen our manuscript

2. Abstract: Clarify the specific objective of exploring interventions, summarize the sub-themes mentioned, and strengthen the practical implications with concrete recommendations for future interventions.

Response: We thank the reviewer for their comment. We have clarified the specific objective, summarized the sub-themes, and strengthened the practical implications with concrete recommendations for future interventions in the abstract as follows:

Abstract

Aim: This study aimed to explore the strategies and important components that can be implemented to build adolescent awareness about stunting prevention.

Methods: This study used descriptive qualitative design. The data were collected through focus group discussions (FGDs) and semi-structured interviews. Purposive sampling method was employed to select the participants. The FGDs involved adolescents (n=6) and high school counselling teachers (n=5), while the semi-structured interviews were conducted with experts frequently involved in overcoming stunting problems in Indonesia (n=7). The interview results were transcribed in verbatim transcription and analysed by using thematic analysis. 

Results: Five themes were identified from the results: 1) Adolescent identity development with three sub-themes: online identity exploration, rebellious stage, and peer influence; 2) Creative and visually appealing website with six sub-themes: interesting appearance, short time span, serial content, story pattern, scenario using adolescent idol’s name, and attractive website menu; 3) Nutritional needs for adolescents with three sub-themes: iron and calcium intake, less sugar consumption, and nutritional status; 4) Engaging content for adolescents with seven sub-themes: stunting, reproductive health, anaemia, diet, wellness, early marriage, and physical activity; and 5) Effective communication strategy with two sub-themes: consistency of activities and communicative. 

Implications: In designing adolescent stunting prevention interventions, multidisciplinary programs utilizing engaging digital health modules and grassroots partnerships should be developed and tested. These programs aim to enhance knowledge retention among youth through appealing online content and interactive community activities. Rigorous evaluation of biopsychosocial approaches can establish integrated best practices across individual, social and policy dimensions for reducing stunting.

3. Introduction: please provide a sharp gap by explicitly stating the specific gap in existing research or interventions your study aims to address.

Response: The authors appreciated the reviewer's comment. We have addressed the sharp gap by clearly identifying the specific gap in existing research in the introduction section as follows:

Introduction

Stunting education for adolescents has been implemented across Indonesia, including programs such as "PENTINGJADI" in West Sumatra [19], educational media in Central Java [20], and a nutrition curriculum in North Sumatra [21]. These aimed to im

---

## [Decision Letter · Decision Letter 2]

14 Nov 2024

Exploring the strategies and components of interventions to build adolescent awareness about stunting prevention in West Java: A qualitative study

PONE-D-24-00216R2

Dear Dr. Nurhaeni,

We’re pleased to inform you that your manuscript has been judged scientifically suitable for publication and will be formally accepted for publication once it meets all outstanding technical requirements.

Kind regards,

Sana Sadiq Sheikh

Academic Editor

PLOS ONE

Additional Editor Comments (optional):

Reviewers' comments:

Reviewer's Responses to Questions

**Comments to the Author**

1. If the authors have adequately addressed your comments raised in a previous round of review and you feel that this manuscript is now acceptable for publication, you may indicate that here to bypass the “Comments to the Author” section, enter your conflict of interest statement in the “Confidential to Editor” section, and submit your "Accept" recommendation.

Reviewer #3: All comments have been addressed

Reviewer #4: All comments have been addressed

2. Is the manuscript technically sound, and do the data support the conclusions?

Reviewer #3: Yes

Reviewer #4: Yes

3. Has the statistical analysis been performed appropriately and rigorously? 

Reviewer #3: N/A

Reviewer #4: Yes

4. Have the authors made all data underlying the findings in their manuscript fully available?

Reviewer #3: Yes

Reviewer #4: Yes

5. Is the manuscript presented in an intelligible fashion and written in standard English?

Reviewer #3: Yes

Reviewer #4: Yes

6. Review Comments to the Author

Reviewer #3: Dear PLOS ONE Editorial Team,

Thank you for the reminder. I apologize for the delay in submitting my review for the manuscript titled, "Exploring the strategies and components of interventions to build adolescent awareness about stunting prevention in West Java: A qualitative study."

I am pleased to report that the authors have thoroughly addressed all the suggestions and comments from the initial review and have made the necessary improvements. The manuscript is now well-structured and effectively presents the study's objectives, methodology, and findings.

Based on these revisions, I believe the manuscript is suitable for publication in its current form.

Thank you for the opportunity to contribute to this review process.

Best regards,

Dr. Suyanto

Reviewer #4: This study provides valuable insights into building adolescent awareness about stunting prevention, though several areas warrant improvement. Ethical approvals and informed consent were appropriately obtained, addressing the study's sensitivity with minors. The limited sample of six adolescents and five teachers reached thematic saturation, yet broader generalizability could be achieved with a larger, more diverse participant pool. The online data collection method extended geographic reach but introduced potential biases in response depth, acknowledged as a study limitation. The authors transparently disclosed funding sources and declared no competing interests. To enhance the study's impact, further comparative analysis with existing Indonesian research on stunting prevention is recommended, along with a discussion on regional variations and specific policy implications. Practical recommendations could include more tailored interventions reflecting Indonesia's cultural and regional diversity to improve adolescent engagement in stunting prevention efforts

7. PLOS authors have the option to publish the peer review history of their article (what does this mean?). If published, this will include your full peer review and any attached files.

Reviewer #3: No

Reviewer #4: No

---

## [Editor Report · Acceptance letter]

25 Nov 2024

PONE-D-24-00216R2 

PLOS ONE

Dear Dr. Nurhaeni, 

I'm pleased to inform you that your manuscript has been deemed suitable for publication in PLOS ONE. Congratulations! Your manuscript is now being handed over to our production team.

Kind regards, 

on behalf of

Dr. Sana Sadiq Sheikh 

Academic Editor

PLOS ONE